# Identification of a New Antimicrobial, Desertomycin H, Utilizing a Modified Crowded Plate Technique

**DOI:** 10.3390/md19080424

**Published:** 2021-07-27

**Authors:** Osama G. Mohamed, Sadaf Dorandish, Rebecca Lindow, Megan Steltz, Ifrah Shoukat, Maira Shoukat, Hussein Chehade, Sara Baghdadi, Madelaine McAlister-Raeburn, Asad Kamal, Dawit Abebe, Khaled Ali, Chelsey Ivy, Maria Antonova, Pamela Schultz, Michael Angell, Daniel Clemans, Timothy Friebe, David Sherman, Anne M. Casper, Paul A. Price, Ashootosh Tripathi

**Affiliations:** 1Natural Products Discovery Core, Life Sciences Institute, University of Michigan, Ann Arbor, MI 48109, USA; ogomaa@umich.edu (O.G.M.); pamschu@umich.edu (P.S.); 2Life Sciences Institute, University of Michigan, Ann Arbor, MI 48109, USA; davidhs@umich.edu; 3Pharmacognosy Department, Faculty of Pharmacy, Cairo University, Kasr el-Aini Street, Cairo 11562, Egypt; 4Biology Department, Eastern Michigan University, Ypsilanti, MI 48197, USA; sdorandi@emich.edu (S.D.); rlindow@emich.edu (R.L.); msteltz@emich.edu (M.S.); ishoukat@emich.edu (I.S.); mshoukat@emich.edu (M.S.); gm0971@wayne.edu (H.C.); sbaghdad@emich.edu (S.B.); mcali22m@mtholyoke.edu (M.M.-R.); akamal3@emich.edu (A.K.); dabebe1@emich.edu (D.A.); kali5@emich.edu (K.A.); ichelsey@emich.edu (C.I.); mantono1@emich.edu (M.A.); mangell@emich.edu (M.A.); dclemans@emich.edu (D.C.); anne.casper@emich.edu (A.M.C.); 5Biological Sciences, Mount Holyoke College, South Hadley, MA 01075, USA; 6Biological Sciences, Wayne State University, Detroit, MI 48202, USA; 7Chemistry Department, Eastern Michigan University, Ypsilanti, MI 48197, USA; tfriebe@emich.edu; 8Department of Medicinal Chemistry, College of Pharmacy, University of Michigan, Ann Arbor, MI 48109, USA

**Keywords:** antibiotic discovery, crowded plate technique, ESKAPE pathogens, natural products, Tiny Earth^TM^

## Abstract

The antibiotic-resistant bacteria-associated infections are a major global healthcare threat. New classes of antimicrobial compounds are urgently needed as the frequency of infections caused by multidrug-resistant microbes continues to rise. Recent metagenomic data have demonstrated that there is still biosynthetic potential encoded in but transcriptionally silent in cultivatable bacterial genomes. However, the culture conditions required to identify and express silent biosynthetic gene clusters that yield natural products with antimicrobial activity are largely unknown. Here, we describe a new antibiotic discovery scheme, dubbed the modified crowded plate technique (mCPT), that utilizes complex microbial interactions to elicit antimicrobial production from otherwise silent biosynthetic gene clusters. Using the mCPT as part of the antibiotic crowdsourcing educational program Tiny Earth^TM^, we isolated over 1400 antibiotic-producing microbes, including 62 showing activity against multidrug-resistant pathogens. The natural product extracts generated from six microbial isolates showed potent activity against vancomycin-intermediate resistant *Staphylococcus aureus*. We utilized a targeted approach that coupled mass spectrometry data with bioactivity, yielding a new macrolactone class of metabolite, desertomycin H. In this study, we successfully demonstrate a concept that significantly increased our ability to quickly and efficiently identify microbes capable of the silent antibiotic production.

## 1. Introduction

Antibiotic resistance is one of the greatest threats to global health, food security, and economic development. In 2017, the World Health Organization (WHO) reaffirmed that antimicrobial resistance is a global health emergency that will seriously jeopardize progress in modern medicine, noting that if antibiotics lose their effectiveness, common medical interventions such as cesarean sections, cancer treatments, and hip replacements will become incredibly risky and transplant medicine will be virtually impossible [1,2,3]. The global rate of infections caused by multidrug-resistant ESKAPE pathogens (*Enterococcus faecium*, *Staphylococcus aureus*, *Klebsiella pneumoniae*, *Acinetobacter baumannii*, *Pseudomonas aeruginosa*, and *Enterobacter* species) has risen dramatically in recent years, resulting in 3–4 million infections and 55,000 deaths annually in the United States and European Union [4,5]. Conservative estimates put the additional cost of these infections at USD 20–35 billion in direct health care costs and 8 million extra days in the hospital each year in the US alone [6,7]. If nothing changes to accelerate the current rate of antimicrobial discovery, future projections suggest that by 2050, 10 million mortalities a year will result from antibiotic-resistant bacteria, leading to a 2%–3.5% reduction in GDP with worldwide costs approaching USD 100 trillion, surpassing the mortality rates for cancer, diabetes, and road traffic accidents [8]. An analysis by the PEW Charitable Trusts indicated that there are currently 42 new antimicrobial compounds in clinical development, only 17 of which are effective against high-priority Gram-negative organisms, only one of which is a novel class of antibiotics. These numbers are far from what is needed to ensure a “robust” drug discovery pipeline in the future [9,10].

Natural products have long provided the scaffolding for most of our current antibiotics. Still, the search for new antibiotics from natural products was abandoned mainly due to the high rate of rediscovery using traditional axenic cultivation techniques [11]. Despite the widely held perception that antibiotic discovery from natural products has reached its commercially viable limits, recent metagenomic data have unequivocally demonstrated that there is still a vast wealth of natural product biosynthetic potential encoded in both cultivatable and uncultivatable bacterial genomes [11,12]. Indeed, the recent announcements of new antibiotic classes characterized by teixobactin and malacidins indicate that non-traditional approaches to natural product discovery (iCHIP and metagenomic-directed screening) can be used to tap the wealth of bioactive compounds that have antimicrobial activity [13,14]. While these approaches have made significant contributions to antibiotic discovery, they largely ignore the potential for silent biosynthetic gene clusters (BGCs) and/or attempt to uncouple BGC expression from the ecological factors that would normally control their production. Since many antibiotics have self-harming effects even with relevant resistance genes, the expression of silent BGCs with antibiotic potential is likely to be tightly controlled. Several chemical and physical signals have been shown to induce and/or increase the production of antimicrobial compounds from otherwise silent BGCs, including nutrient limitation, autoinducers, bacterial and plant cell wall components (e.g., *N*-acetyl glucosamine and plant polysaccharides), phytohormones, other antibiotics, varying media composition, and coculture with various bacteria and/or fungi [15,16,17,18,19,20,21]. However, many studies examining the expression of silent BGCs are frequently limited to screening preestablished culture collections with little or no prior knowledge of a bacterium’s ability to produce antimicrobial compounds. For example, our natural product extracts library housed at the Natural Products Discovery Core, Life Sciences Institute, at the University of Michigan consists of >50 k samples isolated from primarily marine organisms (99%) from around the world, including North America, South America, Asia Pacific, the Middle East, and Antarctica. Our current methodology of microbial isolation is mainly focused on increasing strain diversity and not on the established bioactivity of potential isolates. Moving forward, one of our strategies to aid future drug discovery involves increasing the percentage of natural product extracts with confirmed bioactivity along with microbial diversity isolated from a variety of sources. 

New ecology-based screening methods are therefore needed that can efficiently identify microbes that encode for silent antibiotic production during the initial isolation process. Of the antibiotic-inducing factors mentioned above, microbes’ grown under cocultivation conditions have had good, although limited, success in inducing antibiotic production from otherwise silent BGCs. Cocultivation likely provides physical and chemical stimuli that would typically not be available during axenic growth [16,22]. However, determining which microbes to use as costimulatory partners often requires testing hundreds of pairwise interactions or using a high-throughput coculture system [23]. Some initial antibiotic screening methodologies relied on spontaneous microbial interactions to induce and screen for antibiotic-producing microbes from diluted or undiluted soil samples. These methods include the Waksman’s crowded plate technique [24], Stansley’s and Wilska’s use of sprayers to apply target organisms following microbial growth [25,26], Foster and Woodruff’s incorporation of target organisms into the underlying media [27], and Kelner’s use of agar overlays containing target organisms following microbial growth [28]. However, these techniques and numerous other variants had many drawbacks that ultimately limited their use in large screening operations. For example, the lack of reproducibility in continued antibiotic production under axenic growth conditions following primary isolation. Difficulty in isolating the antibiotic-producer away from the target organism and surrounding microbes. Bias toward fast-growing microorganisms and the requirement of excessive labor [29,30]. Here, we describe here a simple yet improved adaptation of Waksman’s crowded plate technique, which relies on random interactions between densely plated microbes to observe zones of inhibition.

In contrast, our new modified crowded plate technique (mCPT) simultaneously inoculates a decreased number of environmental organisms with an excess of a sensitized target organism allowing us to directly screen complex microbial communities for antibiotic-producing microbes, many of which exhibit silent antibiotic production under axenic growth conditions. Using our developed technique, we isolated over 1400 culturable antibiotic-producing isolates from soil, aquatic, and marine environments as part of the Tiny Earth^TM^ Program [31]. More importantly, we coupled our culture technique with a data-based chemical analysis [32,33], leading to the identification of a new macrocyclic polyketide class of antibiotic, desertomycin H (**2**), together with the known congener desertomycin A (**1**). Structure elucidation of the metabolites was performed based on detailed spectroscopic analysis.

## 2. Results

### 2.1. Waksman’s Crowded Plate Technique

During our initial attempts to establish a simplified screening tool that could efficiently detect antibiotic production within complex microbial communities, we had the most success with Waksman’s crowded plate technique (CPT), which involves plating a high density of microbes on agar media to observe growth inhibition between interacting members of the community (Figure 1A) [30]. Similar to the original reports [28], our attempts to recreate Waksman’s CPT using nutrient-rich agar plates (nutrient agar, 10% TSA, or TY) failed to produce notable zones of inhibition after two weeks. In contrast, when nutrient-limiting agar media (R2A) was used, Waksman’s CPT was more successful at identifying potential antibiotic-producing microbes. However, attempts to confirm antibiotic production from the CPT isolates using Waksman’s “cross streak method” under axenic culture conditions (R2A agar) largely failed to yield antibiotic activity against the antibiotic-sensitive ESKAPE pathogens (2/30 or 6.7%). Therefore, we attempted to recreate a more stimulatory environment by simultaneously patching 16–20 CPT isolates with a target organism spread over the entire plate (Figure 2A,B), also known as the spread-patch method [34]. In this instance, both the target pathogen and neighboring antibiotic producers can provide potential sources of external stimuli that might be needed to activate silent BGCs from culturable organisms. Using the spread-patch method, approximately 25% of antibiotic-producers identified using the CPT displayed antimicrobial activity following purification (Appendix A).

In comparison, a random assortment of 210 culturable microbes selected for screening using the spread-patch method resulted in the isolation of only 13 antibiotic producers, an efficiency rate of just 6.2%. Further streptomycetes’ isolation based on colony morphology resulted in an efficiency rate of 18% using the spread-patch method (Appendix A). Interestingly, when we let the CPT plates incubate over an extended period of time, we noticed the development of two major groups of antibiotic producers: antibiotic producers that inhibited the spread of bacterial species (mainly *Paenibacillus* sp.) over a plate (as seen in Figure 1A) and antibiotic producers that were able to lyse adjacent colonies (often observed as crescent-shaped colonies) (Figure 1B). Upon retesting using the spread-patch plate assay, the majority of the latter group (able to lyse adjacent colonies) retained their activity, comprising most of the 25% of isolates that retained their activity upon retesting (observed data). These observations suggested that additional adjustments to Waksman’s CPT would further increase our ability to identify antibiotic-producing microbes from complex communities.

### 2.2. Modified Crowded Plate Technique (mCPT)

We therefore tested whether co-inoculating diluted sediment samples with various target organisms would result in zones of inhibition when incubated for extended periods. Although no zones of inhibition were observed at one day post-inoculation, we were able to observe small zones of inhibition forming over time (Figure 1C). Given this new method’s similarity to Waksman’s original CPT but with an added target organism, we have designated this method the “modified crowded plate technique (mCPT)”. Like our observations with Waksman’s CPT, media choice was critical in successfully implementing this technique. Nutrient-rich media (10% TSA and TY) produced very few inhibition zones over four weeks. In contrast, nutrient-limiting media (R2A) and the streptomycete sporulation medium MYM slightly increased our ability to identify zones of inhibition over the same four-week period (Appendix A). After seeing these media-dependent differences, we tested several new nutrient-limiting medium formulations for their ability to enhance the effects of the mCPT. One medium formulation in particular, designated TYME, significantly increased our ability to identify soil bacteria that displayed antimicrobial properties. The TYME medium was intended to limit the carbon and nitrogen availability while supplementing essential elements typically provided in trace amounts in other media components. As a result, the TYME medium formulation almost tripled the number of inhibition zones observed over the same four-week period using the same diluted soil samples (Appendix A). Notably, the number of inhibition zones almost doubled from week one to week four post-inoculation when either *Escherichia coli* or *Bacillus subtilis* were used as target organisms. Since the TYME medium is nutrient-limiting and the inhibition zones are relatively small, mCPT plates can then be observed for weeks or even months for new antibiotic producers, limiting the bias toward fast-growing microbes. In some cases, we observed new inhibition zones form four months post-inoculation, mainly from slow-growing bacteria.

### 2.3. d-Alanine Auxotrophs Amplify the Effects of Antibiotic Production and Simplify Purification

Although we extensively used wild-type *E. coli* and *B. subtilis* for early development of the mCPT screening method, *Enterobacter*, *Klebsiella*, *Mycobacterium*, *Pseudomonas*, *Salmonella*, and *Staphylococcus* species were also successfully used to identify antibiotic producers. However, purification of antibiotic producers away from the target organism still limited the use of the mCPT with most of these organisms. We also found that purifying antibiotic producers isolated away from the target organism was extremely time-intensive and repeatedly failed, especially when spore-forming bacteria like *B. subtilis* were used. To alleviate the purification complications due to the target organism, we tested whether auxotrophic mutants could be used for the initial screening process. Specifically, d-alanine auxotrophic mutants were tested because they will only grow on media supplemented with d-alanine, allowing us to create a wide variety of complex yet nutrient-limiting media while still efficiently removing the target organism during the purification process. Additionally, Gram-positive d-alanine auxotrophs are more sensitive to antibiotics as d-alanine is used in both peptidoglycan and wall teichoic acids, leading to a more rapid lysis of affected cells under d-alanine limiting conditions [30]. Further, a direct comparison between prototrophic and d-alanine auxotrophic *B. subtilis* strains using 50 different soil samples showed no notable zones of inhibition on day 1. However, by three months post-inoculation, 346 zones of inhibition were observed for the prototrophic *B. subtilis*, averaging 3.96 mm in size, whereas 677 zones of inhibition were observed for the D-alanine auxotrophic B. subtilis, averaging 7.2 mm in size (Mann–Whitney U test, U = 8.5, n_1_ = 346, n_2_ = 677, *p* < 0.05, one-tailed). Finally, approximately 75% of purified antibiotic-producing isolates selected using the mCPT (TYME medium with d-alanine auxotrophic *B. subtilis*) retained their ability to produce antimicrobial compounds active against *B. subtilis* during secondary testing with the spread-patch assay (Figure 2A and Appendix A). Greater than 85% of antibiotic producers that inhibited *B. subtilis* on the spread-patch assay (Figure 2A) also inhibited *S. aureus* (Figure 2B), confirming the utility of the mCPT method in identifying antibiotic-producing microbes that might have clinical significance. 

### 2.4. Dereplication Using Multidrug (MDR) and Extensively Drug-Resistant (XDR) Clinical Isolates

During the two years of implementing the mCPT, we identified over 1400 antibiotic-producing bacteria. Based on 16S rRNA gene sequencing and morphology characteristics examination, approximately 80% of our antibiotic-producing isolates are actinomycetes, including species from the genera *Arthrobacter*, *Curtobacterium*, *Kocuris*, *Microbacterium*, *Micrococcus*, *Rhodococcus*, and *Streptomyces*. Other isolated genera included *Acinetobacter*, *Aeromonas*, *Bacillus*, *Brevibacillus*, *Bosea*, *Chryseobacterium*, *Cupriavidus*, *Lysinibacillus*, *Lysobacter*, *Mitsuaria*, *Paenibacillus*, *Pedobacter*, *Pseudomonas*, *Pseudoxanthomonas*, *Rheinheimera*, and *Stenotrophomonas*. To confirm the potential of mCPT isolates to produce potentially novel compounds, multidrug (MDR) and extensively drug-resistant (XDR) isolates from the CDC and FDA Antibiotic Resistance (AR) Isolate Bank collection of patient isolates were used as target organisms on the spread-patch assay. In brief, 300 confirmed antibiotic-producing soil isolates were tested, and 152 soil isolates actively inhibited vancomycin-intermediate *S. aureus* (VISA5), and 62 soil isolates actively inhibited XDR carbapenem-resistant Enterobacteriaceae (CRE128 and CRE143), generating zones of inhibition greater than 1 mm (Figure 2C,D, respectively). These data indicate that the mCPT and secondary screening methods can identify antibiotic producers with clinical potential.

### 2.5. Dereplication Using the Antibiotic Resistance Platform (ARP)

Since many known antimicrobial compounds are toxic and not used clinically, resistance to these compounds is unlikely to be present in clinical isolates. Thus, Cox et al. recently developed a robust dereplication platform for many known natural antibiotics using a series of cloned antibiotic resistance genes, which they called the antibiotic resistance platform (ARP) [35]. A core panel of 15 resistance genes, which confer high levels of resistance to known antibiotic classes and account for 70% of the most frequently identified antibiotic classes, has been made available for researchers including genes for resistance to aminoglycosides (*armA*, *rmtB*, and *aph(3”)-Ia*), beta-lactams (*bla_NDM-1_*), aminocyclitols (*aph(9)-Ia*), macrolides (*ermC*), streptogramin A (*vatD*), streptogramin B (*vgb*), streptothricin (*STAT*), tetracyclines (*tet(M)*), chloramphenicols (*CAT*), fosfomycin (*fosA*), rifamycins (*arr*), polymyxins (*mcr-1*), and echinomycins (*uvrA*). However, like our previous results, initial attempts (*n* = 10) to use the ARP panel as originally described using axenic culture conditions failed due to the lack of any antimicrobial activity.

In contrast, when 16–20 antibiotic producers were patched onto a plate together with the ARP strains, we were again able to observe the antimicrobial activity. Using a random assortment of 40 antibiotic-producing strains that initially inhibited *E. coli*, including at least half that inhibited XDR-CREs, we observed that only 7 (17.5%) isolates could be dereplicated using the 15 ARP *E. coli* strains (3 streptothricin, 1 beta-lactam, 1 echinomycin, 1 chloramphenicol, and 1 fosfomycin). To help improve this assay’s high-throughput efficiency, we next tested whether it would be possible to combine all 15 ARP strains into a single assay, which would rapidly identify the production of these classes of antibiotics. Therefore, we combined all 15 ARP *E. coli* strains in equal proportions for a single assay (15X-ARP). A separate drug-sensitive *E. coli* strain was used as a control to test for baseline activity. Testing the same 40 isolates as above, the identical seven antibiotic-producing isolates were dereplicated using the 15X-ARP assay (Figure 3). Using the *E. coli* 15X-ARP combinations and the drug-sensitive control strain, we screened 306 antibiotic-producing isolates that were isolated using the mCPT method with d-alanine auxotrophic *B. subtilis* as a target organism and observed that 197 isolates (64.4%) inhibited the drug-sensitive *E. coli* strain and 166 isolates inhibited the 15X-ARP *E. coli* strain combination, indicating that, on average, only 15.7% (31/197) of our collection produces these commonly encountered anti-Gram-negative antibiotics, far lower than the 70% previously reported by Cox et al. [35]. 

### 2.6. Chemical Analysis of Mixed Fermentation Cultures

One of the major pitfalls of plate-based antibiotic discovery techniques is the inability of potential isolates to produce their antimicrobial compounds during liquid fermentation [36]. However, many groups have shown that mixed-culture fermentation systems can increase or induce antibiotic production from silent biosynthetic gene clusters [15,37,38,39,40,41], most recently using mycolic-acid containing microbes [20,42]. We, therefore, tested fourteen antibiotic-producing strains from a mixture of soil, aquatic, and marine environments that showed anti-Gram-negative activity (*Streptomyces* sp. PAP57, PAP58, PAP60, PAP61, PAP62, PAP117, PAP124, PAP133, PAP143, PAP163, PAP181, and MMR14, *Pseudomonas* sp. PAP165, and *Paenibacillus* sp. PAP203), including nine isolates that significantly inhibited XDR CREs (PAP117, PAP124, PAP133, PAP143, PAP163, PAP165, PAP181, PAP203, and MMR14), for their ability to produce antimicrobial compounds in a single fermentation or mixed-culture fermentation system with three different mycolic-acid producing bacteria: *Corynebacterium glutamicum*, *Mycobacterium smegmatis*, and *Rhodococcus erythropolis*. Natural product chemical extracts (NPEs) from strains PAP57, PAP58, PAP60, PAP61, PAP62, and PAP203 inhibited the growth of drug-sensitive and vancomycin-intermediate resistance strains of *S. aureus* (Table 1), but none of the NPEs either from single or mixed fermentation conditions showed activity against antibiotic-sensitive strains of *E. coli*, *K. pneumoniae*, or *P. aeruginosa*. Only the NPEs from PAP61 and PAP62 showed an increase in antibiotic activity when grown under mixed-culture fermentation conditions compared to single-culture fermentation conditions in the A3M medium (Table 1).

### 2.7. Purification and Identification of Antimicrobial Entities

We routinely used standard (24 h) and extended incubation periods (7 days) to test NPE activity to determine the primary inhibitory activity and relative levels of bacteriostatic activity and spontaneous resistance. Interestingly, the NPEs from the PAP62 and PAP203 mixed-culture fermentation generated an initial zone of inhibition and then formed a secondary zone of inhibition that nearly doubled in size by 72 h, effectively killing a lawn of *S. aureus* (observed for both the drug-sensitive and VISA strains) (Appendix A). The reverse-phase chromatographic fractionation of crude extract of PAP62 mixed-culture fermentation (*M. smegmatis*) revealed fractions 6–8 as the most active with inhibition zones measuring 9–20 mm, with fraction 8 showing the formation of the secondary zone of inhibition. We further identified at least three other strains (EMU101, EMU 133, and YNYX119) from our marine microbial library comparable to each other based on the initial antibacterial activity and mass-spectrometry (MS) based bioactivity profiling, suggesting the presence of the same active entity. We opted for full-scale chemical analysis of fraction 8 obtained from the crude extract of PAP62, based on the potential high yield of active entity, which led to the isolation of two bioactive entities, desertomycin A (**1**) and new analog desertomycin H (**2**) (Figure 4).

Desertomycin A (**1**) was purified as a white amorphous solid with HRESI(+)MS analysis revealing a quasimolecular ion ([M + H]^+^) ion peak at *m/z* 1192.7584, indicative of the molecular formula C_61_H_109_NO_21_ (Δmmu −1.9) requiring eight double bond equivalents (DBE). The ^1^H and HSQC NMR (MeOH-*d*_4_) data for **1** revealed resonances for eight olefinic protons (*δ*_H_ 5.30–6.80), a series of oxymethines (*δ*_H_ 3.40–5.10), a series of fourteen methylenes (*δ*_H_ 1.40–2.95, 3.72, and 3.84), seven secondary methyls (*δ*_H_ 0.78, 0.79, 0.88, 0.94, 0.94, 0.97, and 1.11), two tertiary methyls (*δ*_H_ 1.71 and 1.85), and an anomeric proton (*δ*_H_ 4.83). A literature search for this molecular formula and the ^1^H NMR (DMSO-*d*_6_) data suggested **1** was the known antibacterial desertomycin A. First reported in 1958 from *Streptomyces flavofungini* [43]. The structure elucidation of desertomycin A (**1**) was confirmed by comparing ^13^C NMR (DMSO-*d*_6_) with published data [44] and detailed analysis of 2D NMR data (Appendix A).

Further, a new analog, desertomycin H (**2**) was isolated from the same fraction containing compound **1**. The HRESI(+)MS analysis of **2** revealed a pseudomolecular ([M + Na]^+^) ion peak at 1256.7504 consistent with the molecular formula C_63_H_111_NO_22_ (Δmmu −1.4) requiring nine double bond equivalents (DBE), consistent with an acetyl homologue of **1**. This was confirmed with the ^1^H and HSQC NMR (MeOH-*d*_4_) data for **2** showing spectra closely resembling **1**, with additional resonances attributed to an acetamide moiety (*δ*_H_ 1.92 and *δ*_C_ 22.6). The position of acetamide moiety was established based on (i) HMBC correlation from H-46 (*δ*_H_ 3.15, m) to the acetamide carbonyl 46-NHCOCH_3_ (*δ*_C_ 173.2) and (ii) downfield shift of H_2–_46 (*δ*_H_ 3.15, m) in **2** compared to H_2_-46 in 1 (*δ*_H_ 2.93, m) (Appendix A). The structure elucidation was further supported by the analysis of 2D NMR (MeOH-d_6_) data including HSQC, COSY, and HMBC (Figure 5). Consequently, detailed 1D and 2D NMR based analysis revealed, **2**, as the new metabolite desertomycin H (Table 2 and Appendix A).

Attempts to crystallize desertomycin A (**1**) to elucidate the absolute stereochemistry were not successful. However, the relative stereochemistry of C5–C10 and C21–C38 portions of the desertomycin/oasomycin class was predicted through ghd universal NMR database method, which is further proven via synthesis [45,46,47,48]. Our NMR analysis presents comparable data to the already reported desertomycins and therefore we propose similar stereochemistry as reported by previous groups [46,48].

### 2.8. Antibacterial Evaluation

The antibacterial activity of **1** and **2** were examined using a disk diffusion (Appendix A) and minimal inhibitory concentration (MIC) (Table 3) assays against a panel of bacterial strains. As previously published, the antibacterial activity of desertomycin A and H was restricted to Gram-positive bacteria, although the antibacterial activity of desertomycin H was only observed using the disk diffusion method. 

## 3. Discussion

Several groups have recently argued that the natural products produced by silent BGCs likely play ecological roles within communities [15,38]. Specifically, natural products with antimicrobial activity are thought to either coordinate community functions [39] and/or act as competitive inhibitors in nutrient-limiting environments to secure access to resources [16,22,40]. In either case, many antimicrobial compounds are self-harming even with relevant resistance genes. Their production is therefore tightly controlled, relying on physical and chemical cues and the cell’s nutritional status to activate repressed BGCs [36]. Abrudan et al. (2015) showed that when neighboring bacteria interact under nutrient-limiting conditions, it initiates the production of antimicrobial compounds or significantly increases their production. These studies and recent metagenomic data suggest that the antibiotics discovered from axenically grown microbes were just the “low hanging fruit”. Creating conditions where there is a much more complex interplay between microbes might offer a systematic way to induce silent BGCs with antimicrobial activity. From a practical standpoint, several groups have shown that taking known culture collections and performing microcosm or bipartite interactions can help discover novel classes of antibiotics, like the *Staphylococcus*-specific antibiotic amycomicin [41] and the novel class macrobrevin [23]. We show here that exploiting antibiotics’ ecological roles in a competitive, nutrient-limiting environment is a viable and robust method for identifying antibiotic-producing isolates, even when antibiotic production is suppressed under standard laboratory conditions. Using the modified crowded plate technique (mCPT) with D-alanine auxotrophic bacteria and spread-patch secondary screening methods, we identified over 1400 soil microbial isolates. Importantly, all these isolates exhibited antimicrobial activity, including 62 with potent activity against high-priority XDR Gram-negative clinical isolates. Notably, the sheer simplicity and effectiveness of the method make it a powerful crowdsourcing tool for antibiotic discovery.

Virtually all past and present natural product-based antibiotic discovery methods rely on producing an antimicrobial compound prior to the application of the target organism, ultimately looking for the lack of growth of that target organism. In contrast, the mCPT method and accompanying spread-patch secondary screening methods rely heavily on a key concept: the prolonged exposure of bacteria to both bactericidal and bacteriostatic antibiotics will eventually result (at least 3–4 days) in the formation of zones of inhibition due to the activity of autolysins in bacterial cell walls [49] or direct killing and lysis of non-replicative cells [50]. These studies present two plausible mechanisms for how our new methodology can detect zones of inhibition when surrounded by bacterial lawns. Consistent with these concepts, the zones of inhibition observed on mCPT plates took longer to develop and are much smaller than those reported using agar overlays [28,29]. Importantly, the smaller zones of inhibition allowed us to inoculate a higher density of soil microbes, thus increasing the physical and/or chemical contacts that are likely needed to induce the production of natural products from silent BGCs [16,20,22]. Smaller zones of inhibition also allow for more extended incubation periods (months), thus reducing the bias of post-inoculation methods toward fast-growing microbes. 

In addition to its role in creating a competitive environment, the nutrient-limiting media for the mCPT method (TYME) had other significant benefits. It restricts fast-growing soil microbes’ growth while giving slow-growing microbes time to adapt to growth on agar-based media. Similarly, the target organism’s growth is also restricted, making minor inhibition zones more observable. The combined use of our nutrient-limiting, buffered TYME media and the pre-establishment of the target organism also dramatically reduces the number of zones of inhibition resulting from the production of organic acids in the media, which previously yielded many false positives [29]. Thus, when zones of inhibition are observed around colonies, they are more likely to be target compounds for future research (even if they are known compounds) than compounds that merely inhibit the target organism’s initial growth.

Just as important as our primary mCPT screen, our secondary screening methodology (spread-patch assay) also creates a stimulatory environment for antibiotic production. Several labs have shown that antibiotics can act as signaling molecules to induce antibiotic production in neighboring microbes [16,19,51]. Thus, during our secondary screening, antibiotic producers selected from our mCPT plates were placed relatively close together (16–20 small patches per plate) to allow possible antibiotic crosstalk between antibiotic producers and interactions with the co-inoculated target organism. Since nearly every strain selected for secondary screening had the potential to produce antimicrobial compounds, we could maintain a stimulatory environment where silent BGCs could be expressed. In contrast, using the Waksman’s cross-streak method using axenic cultures to test for antibiotic activity largely failed to reproduce the antibiotic activity we observed from the initial CPT and mCPT screens. 

Furthermore, any new culture methodology will only be effective if the downstream chemical analysis platform is equally efficient. Therefore, we coupled the optimized workflow for awakening silent antibiotic BGCs to the mass-spectrometry (MS) based bioactivity profiling [52]. The approach established a high-throughput platform enabling the identification of a new antibiotic congener, desertomycin H, and the rapid dereplication of the desertomycin A. In addition, these MS/MS spectral data for these compounds are now part of our spectral library for a future global natural products social (GNPS) molecular networking analysis of NPEs, which will further aid in dereplication efforts in the future. Together, these data suggest that the mCPT and secondary screening methods and the improved single- and mixed-culture fermentation conditions coupled with MS technology are more likely to lead to the discovery of novel antibiotics than traditional axenic cultivation and activity-guided purification conditions. However, further optimization of mixed-culture fermentation methods is needed to more consistently express otherwise silent BGCs that produce antimicrobial compounds, especially those with anti-Gram-negative activity, which were not observed in this study even though all 14 strains were examined originally showed anti-Gram-negative activity during primary and/or secondary screening. 

In conclusion, we believe that our mCPT methodology coupled with MS-based bioactivity methodology can transform the search for natural products with antimicrobial activity by mining silent BGCs previously suppressed using traditional cultivation techniques. We were able to use this technique to identify over 1400 soil microbes that produce antimicrobial compounds, including some capable of inhibiting high-priority MDR and XDR ESKAPE pathogens. Not only do these data confirm that cultivatable microbes have the genetic potential to produce a broader array of antibiotic compounds when properly stimulated, our new mCPT methodology provides a new mechanism for efficiently identifying these microorganisms, which is the first step in supplying the antibiotic discovery pipeline with novel classes of antimicrobial compounds and addressing the global health emergency of antibiotic resistance.

## 4. Materials and Methods

### 4.1. General Experimental Details

Chemicals were purchased from Sigma-Aldrich (St. Louis, MO, USA) or Merck (Kenilworth, NJ, USA) unless otherwise specified. Analytical-grade solvents were used for solvent extractions. Solvents used for HPLC, UPLC, and HPLC-MS purposes were of HPLC grade supplied by Labscan or Sigma-Aldrich and filtered/degassed through the 0.45 μm polytetrafluoroethylene (PTFE) membrane prior to use. Deuterated solvents were purchased from Cambridge Isotopes (Tewksbury, MA, USA). 

Semipreparative HPLCs were performed using Shimadzu LC-20AT HPLC instruments (Columbia, MD, USA) with corresponding detectors, fraction collectors, and software inclusively. Electrospray ionization mass spectra (ESIMSs) were acquired using the Shimadzu LC-20AD (Columbia, MD, USA) separations module equipped with the Shimadzu LCMS-2020 (Columbia, MD, USA) Series mass detector in both the positive and negative ion modes under the following conditions (Zorbax Eclipse XDB-C8 5 μm column, 150 mm × 4.6 mm, eluting with 1.0 mL/min of isocratic 90% H_2_O/MeCN for 1 min followed by gradient elution to 100% MeCN (with isocratic 0.1% HCO_2_H modifier) over 15 min, at 210 and 254 nm). UHPLC-QTOF analysis was performed on the UHPLC-QTOF instrument comprising an Agilent (Santa Clara, CA, USA) 1290 Infinity II UHPLC (Phenomenex Kinetex 1.7 μm column, 50 mm × 2.1 mm, eluting with 0.4 mL/min of isocratic 90% H_2_O/MeCN for 1 min followed by gradient elution to 100% MeCN over 6 min (with isocratic 0.1% formic acid modifier)) coupled to an Agilent 6545 LC/Q-TOF-MS system operating in the positive mode, monitoring a mass range of 100–2000 amu.

NMR spectra were obtained on an Agilent 600 NMR spectrometer (1H: 600 MHz, 13C: 150 MHz) equipped with a 5 mm DB AUTOX PFG broadband probe and a Varian NMR System console, or a Bruker 800 NMR spectrometer (1H: 800 MHz, 13C: 200 MHz) with an Ascend magnet, a Bruker NEO console and equipped with a 5 mm Triple resonance inverse detection TCI cryoprobe with automatic tuning and matching in the solvents indicated and referenced to residual signals (*δ*_H_ 3.31 and *δ*_C_ 49.0 ppm for MeOH, *δ*_H_ 2.50 and *δ*_C_ 39.52 ppm for DMSO) in deuterated solvents. All data analysis was performed using MestReNova NMR software (Version No 14.0.1-23559., Mestrelab research, Escondido, CA, USA).

### 4.2. Bacterial Strains

The bacterial strains used in this study are described in Table 4. Briefly, ESKAPE “safe” relatives (BSL-1) were obtained from the Tiny Earth^TM^, student sourcing antibiotic discovery, culture collection. Drug-sensitive ESKAPE pathogen strains were obtained from the ATCC. Multidrug (MDR) and extensively-drug resistant (XDR) isolates were obtained from the CDC and FDA Antibiotic Resistance (AR) Isolate Bank collection of patient isolates (https://www.cdc.gov/drugresistance/resistance-bank/index.html (accessed on 19 July 2017). David Sherman (University of Michigan) and Miriam Braunstein (University of North Carolina at Chapel Hill) provided us with the mycolic-acid-producing strains. Petra Levin (Washington University in St. Louis) provided us with the *B. subtilis* d-alanine auxotroph developed in the laboratory of Alan Grossman (Massachusetts Institute of Technology). The *E. coli* Stock Center provided us the *E. coli* D-alanine auxotroph developed in the laboratory of Michael Benedik (Texas A&M University).

### 4.3. Reagents, Soil Samples, and Culture Conditions

All chemicals, unless otherwise noted, were obtained from Sigma-Aldrich (St. Louis, MO, USA) or VWR (Radnor, PA, USA). Bacterial culture reagents were obtained from ForMedium LTD (Hunstanton, England) or Becton, Dickinson, and Company (Sparks, MD, USA). Soil samples were collected from various locations around Southeast Michigan, USA including the campus of Eastern Michigan University in Ypsilanti, Michigan, USA. The following media formulations were used (solid media were prepared with 12 g/L agar): MYM (4 g/L maltose, 4 g/L yeast extract, and 10 g/L malt extract), R2A (premix, DOT Scientific, Burton, MI, USA), 10% TSA (premix at 10% of the amount indicated and supplemented to 12 g/L agar, DOT Scientific, Burton, MI, USA), TY (0.5 g/L CaCl_2_·2H_2_O, 3 g/L yeast extract, and 6 g/L tryptone), TYME (0.5 g/L dextrose, 0.5 g/L peptone, 0.5 g/L yeast extract, 0.5 g/L tryptone, 0.5 mM KH_2_PO_4_ (pH 7.0), 0.25 mM MgSO_4_, 0.25 mM CaCl_2_·2H_2_O, and 1 mL/L minor salts solution (1000× minor salts (per liter) = 9.5 g of Na_2_-EDTA·2H_2_O, 7 g of FeSO_4_·7H_2_O, 1 g of H_3_BO_3_, 250 mg of MnSO_4_·H_2_O, 50 mg of ZnSO_4_·7H_2_O, 50 mg of Na_2_MoO_4_·2H_2_O, 50 mg of CuSO_4_, and 10 mg of CoCl_2_)), EPSM (5 g/L potato starch, 0.5 g/L peptone, 0.5 g/L yeast extract, 0.5 g/L tryptone, 0.5 mM KH_2_PO_4_ (pH 7.0), 0.25 mM MgSO_4_, 0.25 mM CaCl_2_·2H_2_O, and 1 mL/L minor salts solution), A3M (5 g/L dextrose, 5 g/L soluble starch, 3 g/L yeast extract, 2 g/L PharmaMedia (ADM, Dacatur, IL, USA), and 2% glycerol), ISP2 (4 g/L yeast extract, 10 g/L malt extract, and 10 g/L dextrose), ISP3 (15 g/L oatmeal flour), and OPAH (1 g/L oatmeal, 1 g/L PharmaMedia, 1 g/L L-arabinose, 0.5 g/L humic acids, 0.5 mM KH_2_PO_4_ (pH 7.0), 0.25 mM MgSO_4_, 0.25 mM CaCl_2_·2H_2_O, and 1 mL/L minor salts solution). All bacterial cultures were grown at room temperature, 30 °C, or 37 °C (for chemical extract tests only) and supplemented as needed: 20 μg/mL natamycin (NataMax SF, DuPont-Danisco USA Inc., New Century, KS, USA) and 100 μg/mL D-alanine. For long-term storage, all isolates were resuspended in 20% glycerol and placed at −80 °C. *Streptomyces* spore stocks were generated on TYME, ISP3, or OPAH agar medium and also stored at −80 °C.

### 4.4. Crowded Plate Technique

Aquatic, marine, and/or soil samples were collected from various locations around Southeast Michigan including the marine ecology labs at Eastern Michigan University. Samples were serially diluted to achieve a total final dilution of 1 × 10^−3^ plating 100 μL on various media. Plates were then grown at 30 °C for 1 week and then maintained at ambient temperature (22 °C) for up to 4 months in covered plastic containers to prevent plates from desiccating. Plates were observed weekly starting on day 1 for zones of inhibition including any signs that the growth of neighboring bacteria was inhibited. Colonies at the geographic center of zones of inhibition were partially purified using a three-streak method and tested for activity using the spread-patch assay. If activity was retained, samples were colony purified and retested for antimicrobial activity using the spread-patch assay.

### 4.5. Modified Crowded Plate Technique

Petri plates were preinoculated with the target organism using a cotton swab from stationary cultures on agar plates. Aquatic, marine and/or soil samples were then serially diluted and 100 μL was plated on various media to achieve a total final dilution of 1 × 10^−5^ or approximately 1000–2000 colonies/plate immediately following the inoculation of the target organism. Plates were then grown at 30 °C for 1 week and then maintained at ambient temperature (22 °C) for up to 4 months in covered plastic containers to prevent desiccation. Plates were observed weekly starting on day 1 for zones of inhibition including any signs that the growth of neighboring bacteria was inhibited. Colonies at the geographic center of zones of inhibition were partially purified using a three-streak method and tested for activity using the spread-patch assay. If activity was retained, samples were colony purified and retested for antimicrobial activity using the spread-patch assay.

### 4.6. Spread-Patch Assays

Petri plates were preinoculated with the target organism using a cotton swab. Potential antibiotic producers were then transferred onto TYME or EPSM agar plates to create small patches (6–10 mm). In total, 16–20 patches of different antibiotic producers were tested on a single plate. Plates were incubated at 30 °C for seven days. Observations were taken at day 1, day 3, and day 7 for zones of inhibition. Due to the various patch sizes, zones of inhibition were measured from the edge of a patch to the end of the zone of inhibition. Images of plates were taken on day 7.

### 4.7. 16S rRNA Gene Sequencing and Analysis

Bacterial isolates were streaked onto TYME plates and cultured for 5 days at 30 °C to use as a template for PCR. Colony PCR using oligonucleotides targeting the 16S rRNA gene (27FHT forward oligonucleotide: 5′- AGR GTT TGA TCM TGG CTC AG -3′; 1492RHT reverse oligonucleotide: 5′- GGY TAC CTT GTT AYG ACT T -3′) were used to amplify and sequence a 1465 bp region. PCR conditions were as follows: 94 °C denaturing for 10 m, 35 cycles (94 °C denaturing for 10 s, 50 °C annealing for 30 s, and elongation at 72 °C for 120 s), and 72 °C final extension for 6 m. Sanger sequencing was performed at the University of Michigan DNA sequencing core (https://brcf.medicine.umich.edu/cores/dna-sequencing/, accessed on 19 July 2017) or Eton Biosciences, Inc. (Union, NJ, USA) using the 27FHT primer. Approximately, 700 bp of the 16S rRNA gene were used to perform a nucleotide BLAST against the NCBI 16S ribosomal rRNA sequence database to identify the closest relative to the genus level. 

### 4.8. Chemical Extractions and Testing

Bacterial isolates were grown on TYME media for 3–5 days before inoculating a 5 mL culture of ISP2 (*Streptomyces*) or TYME (all other strains). Cultures were then grown at 30 °C at 300 rpm for 5 days before 1 mL of culture was back diluted into 100 mL of A3M (*Streptomyces*), TYME, or other media derivatives as described in 250-mL Erlenmeyer flasks. Flasks were grown at ambient temperature (22 °C) at 200 rpm for 7 days. For mixed-culture fermentation experiments, the cocultures (*M. smegmatis*, *C. glutamicum*, or *R. erythropolis*) were grown separately in 5-mL TYME cultures at 30 °C at 300 rpm for 3 days and 1 mL added to the flasks on day 3. For chemical extractions, 1.5 g of amberlite XAD16N resin sealed in a semiporous membrane was used to harvest compounds from the liquid media. Amberlite resin was added on day 7 for single-culture fermentation or day 4 for mixed-culture fermentation. For large-scale fermentation, 10 mL of each culture was back-diluted into 1 L of A3M medium and 20 g of amberlite XAD16N resin was used for the extraction. A 1:1 methanol: ethyl acetate mixture was then used to extract compounds from the amberlite resin. The resulting mixture was dried down in a SpeedVac (Thermo Fisher Scientific, Waltham, MA, USA) and resuspended in 15 mg/mL of DMSO. 

### 4.9. Activity-Guided Fractionation, Purification, and Structural Analysis

*Streptomyces* sp. PAP62 was isolated from soil collected on the campus of Eastern Michigan University as part of the Tiny Earth^TM^ Educational Program. *Streptomyces* sp. PAP62 was grown on TYME agar plates (0.5 g/L dextrose, 0.5 g/L peptone, 0.5 g/L yeast extract, 0.5 g/L tryptone, 0.5 mM KH_2_PO_4_ (pH 7.0), 0.25 mM MgSO_4_, 0.25 mM CaCl_2_·2H_2_O, 1 mL/L of a minor salts solution (1000 × minor salts (per liter) = 9.5 g of Na_2_-EDTA·2H_2_O, 7 g of FeSO_4_·7H_2_O, 1 g of H_3_BO_3_, 250 mg of MnSO_4_·H_2_O, 50 mg of ZnSO_4_·7H_2_O, 50 mg of Na_2_MoO_4_·2H_2_O, 50 mg of CuSO_4_, and 10 mg of CoCl_2_), and 12 g/L agar) for 7 days. Culture tubes containing 10 mL of ISP-2 medium were then inoculated with *Streptomyces* sp. PAP62 sporulating colonies and grown for 3 days at 300 rpm at 30 °C. Finally, 10 mL of culture were transferred to 1 L of A3M media (5 g/L dextrose, 5 g/L soluble starch, 3 g/L yeast extract, 2 g/L PharmaMedia (ADM, Dacatur, IL, USA), and 2% glycerol) in a 2800-mL Fernbach flask (no baffles) and grown for 7 days at 200 rpm at room temperature (22 °C). On day three, *M. smegmatis* cultured in TYME media for 3 days was added to the media. On day four, 20 g of sterilized amberlite XAD16N resin sealed in a semiporous membrane was added to each flask. On day 7, the amberlite XAD16N resin was extracted with 200 mL of 1:1 methanol:ethyl acetate mixture and dried in a RotoVap. The crude extract of *Streptomyces* sp. PAP62 was subjected to flash chromatographic fractionation using an Isolera Selekt (Biotage^®^, Charlotte, NC, USA) utilizing a prepacked Phenomenex^®^ reversed-phase C18 column (40 g). Material was eluted with a flow rate of 50 mL min^−1^ collecting 120 mL fractions. Material was eluted using a 3-solvent gradient system, consisting of H_2_O (solvent A), methanol (solvent B), and acetonitrile (solvent C). The column was first washed with 10% methanol in H_2_O for 1 CV, followed by a linear increasing gradient from 10% to 100% methanol over 12 CV. An isocratic gradient of 100% methanol was then applied for 5 CV, followed finally by an additional isocratic gradient of 100% acetonitrile for 5 CV to yield eight fractions (F1–F8). Fractions were dried into preweighed vials using a V10-touch evaporator (Biotage^®^) coupled with a Gilson GX-271 Liquid Handler (Biotage^®^, Charlotte, NC, USA). 

The final purification of F8 (58 mg) was subjected to chromatographic purification using a semipreparative reversed-phase HPLC (Phenomenex Luna-C_18_, 10 mm × 250 mm, 5 µm, 4 mL/min, isocratic elution with 80% H_2_O/MeCN for 2 min followed by gradient elution from 80% H_2_O/MeCN to 45% H_2_O/MeCN over 48 min including 0.1% formic acid as a modifier) to yield pure compounds desertomycin A (**1**) (3.4 mg) and desertomycin H (**2**) (1.3 mg).

Desertomycin A **(1):** off-white solid; 1D and 2D NMR (800 MHz, methanol-*d_4_*) see Appendix A; HRESI(+)MS m/z 1192.7584 [M + H]^+^ (calcd for C_61_H_110_N_1_O_21_ 1192.7565).

Desertomycin H **(2):** off-white solid; 1D and 2D NMR (800 MHz, methanol-*d_4_*) see Appendix A; HRESI(+)MS m/z 1256.7504 [M + Na]^+^ (calcd for C_63_H_111_NaN_1_O_22_ 1256.7490).

### 4.10. Antibacterial Assay

For activity testing using the disk diffusion method, the target organism was inoculated onto TYME agar plates using cotton swabs; 6-mm cellulose disks were placed on the agar and 10 μL of extract (150 μg total) was applied to the disks. For fractionated or purified samples, 30 μg of sample was added to 6-mm cellulose disks. The agar plates were then incubated at 37 °C for 24 h and 7 days to observe zones of inhibition (diameter measured in millimeters).

For activity testing using minimal inhibitory concentrations (MICs), the EUCAST standard protocols were followed for 96-well plates. Briefly, 2-fold serial dilutions of desertomycin A or H were performed in Mueller–Hinton broth starting at 128 μg/mL. Then, 5 × 10^5^ CFU/mL of each target organism was added to each well for a total volume of 100 μL, incubated at 37 °C for 24 h, and observed for bacterial growth. The well with the lowest antibiotic concentration with no bacterial growth was considered the MIC for that antibiotic. Kanamycin was used as a positive control and DMSO as a negative control for all assays. 

## Figures and Tables

**Figure 1 marinedrugs-19-00424-f001:**
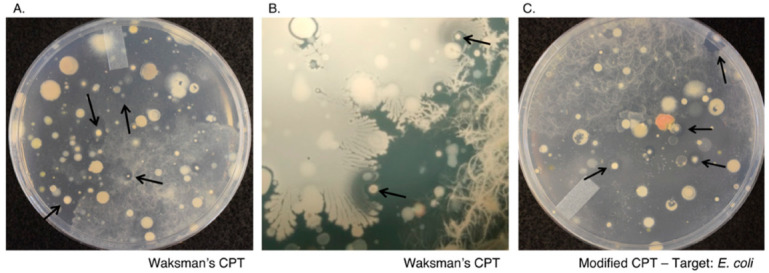
Traditional crowded plate technique (CPT) and modified crowded plate technique (mCPT) on nutrient-poor media. (**A**,**B**) Representative samples of Waksman’s CPT. Antibiotic-producing bacteria (arrows) can be identified by the zones of inhibition surrounding individual colonies following two weeks of incubation at 30 °C on TYME media. (**B**) Antibiotic-producing bacteria (arrows) surrounded by lysed cells forming a zone of inhibition. (**C**) Modified crowded plate technique using a d-alanine auxotrophic strain of *E. coli* pre-inoculated onto the entire plate as a target organism. A diluted soil sample is then immediately spread on the plate. Antibiotic-producing bacteria (arrows) can be identified by the zones of inhibition surrounding individual colonies following two weeks of incubation at 30 °C on TYME media.

**Figure 2 marinedrugs-19-00424-f002:**
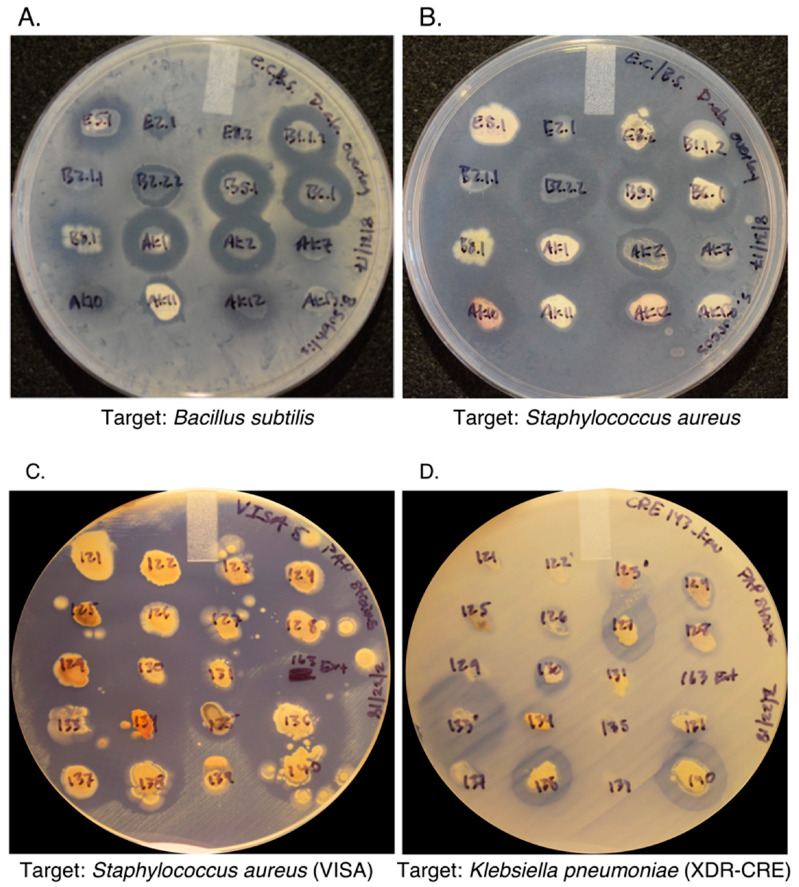
Spread-patch assay confirmation testing of purified antibiotic producers isolated using the modified crowded plate technique (mCPT). (**A**,**B**) Isolates identified using a d-alanine auxotroph of *B. subtilis* were retested on drug-sensitive *B. subtilis* (**A**) and *S. aureus* (**B**). (**C**,**D**) Confirmed antibiotic producers were tested using the spread-patch assay with vancomycin-intermediate strain of *S. aureus* (VISA5—resistant to 9 clinically used antibiotics) (**C**) and a carbapenem-resistant strain of *K. pneumoniae* (CRE143—resistant to 24 clinically used antibiotics) as target organisms (**D**). TYME plates were imaged at 7 days post-inoculation. For most isolates, zones of inhibition on the spread-patch assay are normally not observed until 3 days post-inoculation.

**Figure 3 marinedrugs-19-00424-f003:**
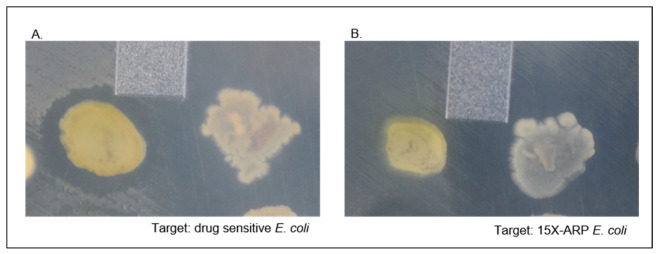
Spread-patch assay confirmation testing of purified antibiotic producers using drug-sensitive *E. coli* or all 15 combined Antibiotic Resistance Panel (15X-ARP) *E. coli* strains. (**A**) Isolates PAP100 (left) and PAP117 (right) were tested on drug-sensitive E. coli with both showing clear zones of inhibition. (**B**) Isolates PAP100 (left) and PAP117 (right) were tested on 15X-ARP *E. coli* strains with PAP100 showing no zone of inhibition and PAP117 showing a large zone of inhibition. For PAP100, the zone of inhibition was also lost when E. coli carrying only the bla_NDM-1_ was used in the spread-patch assay. EPSM plates were incubated at 30 °C and imaged at 7 days post-inoculation.

**Figure 4 marinedrugs-19-00424-f004:**
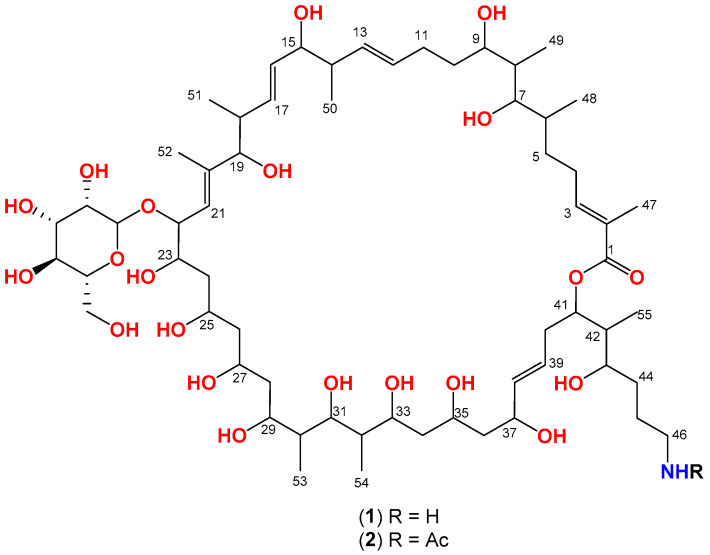
Desertomycins A (**1**) and H (**2**) isolated from *Streptomyces* sp. PAP62.

**Figure 5 marinedrugs-19-00424-f005:**
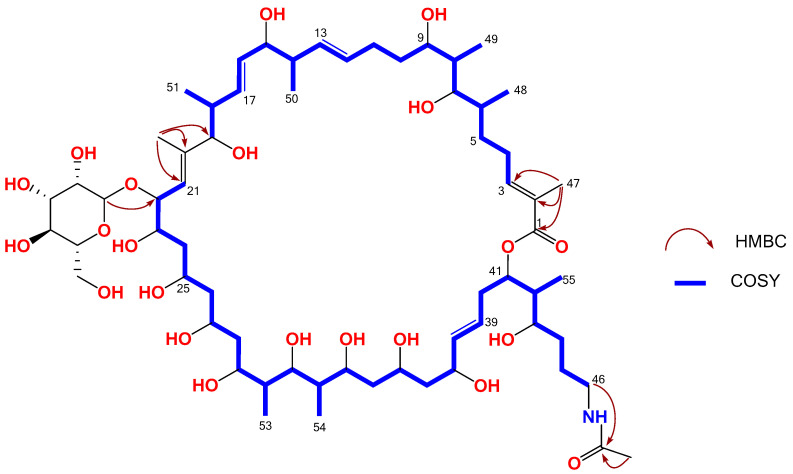
Key 2D NMR correlations of desertomycins H (**2**).

**Table 1 marinedrugs-19-00424-t001:** Effectiveness of chemical extracts against drug sensitive and vancomycin-intermediate resistance *S. aureus* (VISA). Representative experiment of the size (mm) of the zones of inhibition (diameter) observed at 24 h post-inoculation using 150 μg of natural product chemical extracts. Strains were grown in A3M medium alone or with *C. glutamicum* (A3M-Corny), *M. smegmatis* (A3M-Myco), or *R. erythropolis* (A3M-Rhodo).

	Media + Coculture	Target Organism	
Strain		*S. aureus*	VISA
PAP57	A3M	8	7
	A3M-Corny	9	8
	A3M-Myco	8	8
	A3M-Rhodo	9	8
PAP58	A3M	15	18
	A3M-Corny	16	18
	A3M-Myco	12	10
	A3M-Rhodo	13	11
PAP60	A3M	15	12
	A3M-Corny	13	10
	A3M-Myco	13	12
	A3M-Rhodo	13	13
PAP61	A3M	9	6
	A3M-Corny	9	11
	A3M-Myco	10	12
	A3M-Rhodo	8	11
PAP62	A3M	10	9
	A3M-Corny	13	14
	A3M-Myco	12	13
	A3M-Rhodo	14	15
PAP203	A3M	10	9
	A3M-Corny	11	10
	A3M-Myco	10	9
	A3M-Rhodo	10	8

**Table 2 marinedrugs-19-00424-t002:** 1D NMR (MeOH-*d*_4_, 800 MHz) data for desertomycins A (**1**) and H (**2**).

Pos.	1		2		Pos.	1		2	
	*δ*_H_, m (J Hz)	*δ* _C_	*δ*_H_, m (*J* Hz)	*δ* _C_		*δ*_H_, m (J Hz)	*δ* _C_	*δ*_H_, m (J Hz)	*δ* _C_
1		169.3		169.2	33	4.17, td (10.0, 2.3)	70.2	4.17, td (10.0, 2.0)	70.2
2		128.8		128.9	34	1.69 ^b^	43.2	1.70 ^b^	43.2
3	6.80, m	144.4	6.79, m	144.2	35	4.02, m	66.5	4.01, m	66.4
4	2.26 ^a^, m	27.6	2.25 ^a^, m	27.6	36	1.53 ^g^	46.4	1.52 ^f^	46.3 ^l^
5	a 1.42, m	34.5	a 1.41, m	34.4 ^i^	37	4.27, m	69.7	4.27, m	69.7
	b 1.57, m		b 1.56, m		38	5.58, dd (15.4, 5.5)	138.2	5.56, dd (15.5, 5.7)	138.0
6	1.67 ^b^, m	36.0	1.65 ^b^, m	35.9	39	5.62, dd (15.4, 6.7)	125.8	5.62, dd (15.0, 7.2)	125.9
7	3.40, dd (9.6, 1.7)	77.5	3.39, dd (9.6, 1.6)	77.5	40	a 2.30, dd (13.4, 6.7)	34.5	a 2.29, dd (13.5, 7.2)	34.4 ^i^
8	1.75, m	42.8	1.74, m	42.8		b 2.45, dd (13.4, 5.3)		b 2.44, dd (13.5, 5.1)	
9	3.80 ^c^	74.7 ^k^	3.80 ^c^	74.7 ^j^	41	5.10, m	75.6 ^l^	5.11, m	75.7 ^k^
10	a 1.40 ^d^		a 1.38	33.5	42	1.99, q (6.9)	43.7	1.97, q (6.8)	43.5
	b 1.60 ^e^		b 1.59 ^d^		43	3.52, ddd (9.8, 5.6, 1.9)	72.6	3.50, ddd (8.4, 5.5, 1.8)	72.7
11	a 2.07, m	30.40	a 2.07, m	30.4	44	a 1.40 ^d^	30.44	a 1.31	31.0
	b 2.24 ^a^		b 2.24 ^a^			b 1.63 ^e^		b 1.54 ^f^	
12	5.49 ^f^	131.6	5.49 ^e^	131.6	45	a 1.67 ^b^	25.6	a 1.48 ^f^	27.1
13	5.45 ^f^	134.0	5.46 ^e^	134.0		b 1.82, m		b 1.68 ^b^	
14	2.19, q (6.7)	44.1	2.19, q (6.6)	44.0	46	2.93, m	40.8	3.15, m	40.4
15	3.87, dd (6.7, 5.5)	76.7	3.87, dd (6.6, 5.3)	76.7	46-NHCOCH_3_	---	---	---	173.2
16	5.49 ^f^	132.1	5.46 ^e^	132.1	46-NHCOCH_3_	---	---	1.92, s	22.6
17	5.50 ^f^	134.7	5.49 ^e^	134.6	47	1.85, s	12.7	1.85, s	12.7
18	2.34, m	41.2	2.33, m	41.2	48	0.88, d (6.8)	12.6	0.88, d (6.8)	12.5
19	3.72, d (8.5)	83.4	3.71, d	83.4	49	0.78, d (6.9)	12.0	0.78, d (7.5)	12.0
20	---	145.5	---	145.6	50	0.97, d (6.8)	16.1	0.98, d (6.9)	16.1
21	5.30, d (9.6)	124.1	5.30, d (9.3)	124.0	51	1.11, d (6.6)	17.6	1.11, d (6.6)	17.6
22	4.39, dd (9.6, 3.5)	75.6 ^l^	4.39, dd (9.6, 3.6)	75.7 ^k^	52	1.71, s	12.2	1.71, s	12.2
23	3.96, dd (8.3, 3.5)	71.7	3.96, dd (8.0, 3.4)	71.7	53	0.94 ^h^, d (6.9)	10.1	0.94, d (6.9)	10.1
24	1.49, m	41.5	1.49, m	41.46	54	0.79, d (7.0)	11.5	0.79, d (7.5)	11.5
25	4.08, m	66.0	4.08, m	66.0	55	0.94 ^h^, d (6.9)	10.6	0.92, d (6.9)	10.8
26	a 1.53 ^g^	46.4	a 1.53 ^f^	46.3 ^l^	1′	4.83, d (1.5)	97.8	4.83, d (1.3)	97.8
	b 1.62 ^e^		b 1.62 ^d^		2′	3.76 ^i^	72.4 *	3.77 ^g^	72.4 *
27	4.06, m	69.1	4.07, m	69.1	3′	3.76 ^i^	72.5 *	3.75 ^g^	72.6 *
28	1.44, m	42.5	1.44, m	42.55	4′	3.63 ^j^	68.7	3.63 ^h^	68.7
29	3.80 ^c^	75.1	3.80 ^c^	75.1	5′	3.63 ^j^	74.7 ^k^	3.63 ^h^	74.7 ^j^
30	1.63 ^e^	40.8			6′	a 3.72, dd (11.5, 3.6)	62.9	a 3.72	62.9
31	3.98, dd (9.5, 1.7)	73.5	3.98, dd (9.9, 1.7)	73.5		b 3.84, dd (11.5, 1.3)		b 3.84	
32	1.69 ^b^	41.75	1.68 ^b^	41.7					

^a–l^ Overlapping signals within the same superscript letter for the same metabolite and * interchangeable assignment for the same metabolite.

**Table 3 marinedrugs-19-00424-t003:** Antibacterial activity (MIC, μg/mL) of desertomycin A and desertomycin H using EUCAST clinical bacterial standards.

Strain	Desertomycin A (1)	Desertomycin H (2)
*Escherichia coli*	>128	>128
*Enterococcus faecalis*	64	>128
*Mycobacterium luteus*	16	>128
*Staphylococcus aureus* (MSSA) ^1^	32	>128
*Staphylococcus aureus* (VISA5) ^2^	64	>128

^1^ Methicillin-sensitive *Staphylococcus aureus* and ^2^ vancomycin intermediate-sensitive *Staphylococcus aureus*.

**Table 4 marinedrugs-19-00424-t004:** Bacterial strains used in this study.

Strain	Genotype	Source/Reference
PP655	*Bacillus subtilis* 168 *dal-1 sigB::erm* (from AG232 [53])	Alan Grossman
PP662	*Escherichia coli* F-, *Δ(araA-leu)7697*, *[araD139]_B/r_*, *Δ(codB-lacI)3*, *galK16*, *galE15*(GalS), *λ-*, *e14-*, *dadX100::FRT*, *relA1*, *rpsL150*(strR), *spoT1*, *alr-100::FRT*, *mcrB1*	*E. coli* Stock Center [54]
PP663	*Cornybacterium glutamicum* ATCC 13869	David Sherman
PP664	*Rhodococcus erythropolis* B-16025.	David Sherman
PP665	*Klebsiella pneumoniae* ATCC 13883	ATCC
PP666	*Pseudomonas aeruginosa* ATCC 27853	ATCC
PP667	*Staphylococcus aureus* ATCC 25923	ATCC
PP673	*Mycobacterium smegmatis* MC2155	Miriam Braunstein
PP740	*Escherichia coli* DH5α pGDP3_aph(3”)-Ia	AddGene
PP741	*Escherichia coli* DH5α pGDP3_rmtB	AddGene
PP742	*Escherichia coli* DH5α pGDP3_apmA	AddGene
PP743	*Escherichia coli* DH5α pGDP3_aph(9)-Ia	AddGene
PP744	*Escherichia coli* DH5α pGDP3_NDM-1	AddGene
PP745	*Escherichia coli* DH5α pGDP3_ermC	AddGene
PP746	*Escherichia coli* DH5α pGDP3_vatD	AddGene
PP747	*Escherichia coli* DH5α pGDP3_stat	AddGene
PP748	*Escherichia coli* DH5α pGDP3_tet(A)	AddGene
PP749	*Escherichia coli* DH5α pGDP3_cat	AddGene
PP750	*Escherichia coli* DH5α pGDP3_fosA	AddGene
PP751	*Escherichia coli* DH5α pGDP3_arr	AddGene
PP752	*Escherichia coli* DH5α pGDP3_uvrA	AddGene
PP753	*Escherichia coli* DH5α pGDP3_vph	AddGene
PP754	*Escherichia coli* DH5α pGDP3_MCR-1	AddGene
PP771	*Enterococcus faecalis* ATCC29212	ATCC
PP788	*Micrococcus luteus*	This study
PAP57	*Streptomyces* sp. PAP57	This study
PAP58	*Streptomyces* sp. PAP58	This study
PAP60	*Streptomyces* sp. PAP60	This study
PAP61	*Streptomyces* sp. PAP61	This study
PAP62	*Streptomyces* sp. PAP62	This study
PAP117	*Streptomyces* sp. PAP117	This study
PAP124	*Streptomyces* sp. PAP124	This study
PAP133	*Streptomyces* sp. PAP133	This study
PAP143	*Streptomyces* sp. PAP143	This study
PAP163	*Streptomyces* sp. PAP163	This study
PAP165	*Pseudomonas* sp. PAP165	This study
PAP181	*Streptomyces* sp. PAP181	This study
PAP203	*Paenibacillus* sp. PAP203	This study
MMR14	*Streptomyces* sp. MMR14	This study
TE-Ec	*Escherichia coli* ATCC 1775	Tiny Earth^TM^
TE-Bs	*Bacillus subtilis*	Tiny Earth^TM^
VISA5	Vancomycin-intermediate resistant *Staphylococcus aureus*	AR Bank
CRE128	Carbapenem-resistant *Escherichia coli*	AR Bank
CRE143	Carbapenem-resistant *Klebsiella pneumoniae*	AR Bank

## Data Availability

Data is contained within the article or Appendix A.

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
