# Peer review of "Identification of a New Antimicrobial, Desertomycin H, Utilizing a Modified Crowded Plate Technique"

_marinedrugs, 2021, doi:10.3390/md19080424_

Round 1

Reviewer 1 Report

The authors reported a modified crowded plate technique (mCPT) and demonstrated its application in the discovery of a new analog antimicrobial. The so-called mCPT technique coupled with downstream mass spectrometry showed an integrated solution and great potential for exploring new antibiotics in the future. Although no new classes of antibiotics have been found using their proposed scheme and the writing style of the manuscript may have to be improved, I support a publication of their work after a minor revision. 

First of all, I strongly suggest the authors give some concise words and a figure representation regarding the difference between traditional CPT and the modified CPT in earlier sections of the manuscript. This would help the readers to quickly catch the most important argument of this work. Then, the figure or sub-figures should be reorganized or combined. Some figures in the supplementary information can be moved to the main body as I felt no differences between the main figures and supplementary figures in terms of importance. The table information can also be reorganized or simplified. 

There were many “data not shown” in the manuscript. I am not sure if the journal allows “data not shown.” If the answer is no, I suggest the authors offer the relevant data in the supplementary information as far as possible. 

The funding support declaration may be revised to include the financial support from Eastern Michigan University instead of acknowledging it only in the Acknowledgments. Here should follow the journal guideline. 

Reviewer 2 Report

Manuscript ID: marinedrugs-1309955 

Title: Identification of a New Antimicrobial, Desertomycin H, Utilizing a Modified Crowded Plate Technique  

Authors: Osama G. Mohamed, Sadaf Dorandish, Rebecca Lindow, Megan Steltz, Ifrah Shoukat, Maira Shoukat, Hussein Chehade, Sara Baghdadi, Madelaine McAlister-Raeburn, Asad Kamal, Dawit Abebe, Khaled Ali, Chelsey Ivy, Maria Antonova, Pamela Schultz, Michael Angell, Daniel Clemans, Timothy Friebe, David Sherman, Anne M. Casper, Paul A. Price *, Ashootosh Tripathi *

This manuscript describes the new methodology to screen the potential antibiotic-producing strains from the natural environment, which are generally silent in their antibiotic production once isolated and purified. Recent metagenomic analyses have revealed the potential biosynthetic gene clusters of antibiotics, but most gene clusters are not expressed. The physiological functions of these antibiotics and the molecular mechanisms of their expression are still a hot target subject of research. 

The involvement of specific species interactions in the expression of antibiotic synthesis gene clusters has also been pointed out. However, identifying these species is a factor that makes the search for new antibiotics difficult. Therefore, the authors improved on the traditional method introduced by Waksman in 1945 and screened from soil bacterial provided by an undergraduate student project, Tiny Earth.

I think this manuscript has potential interest for the readers of this journal.

Before publication, these criticisms and comments below should be cleared,

1) As for the authors, I believe that the Tiny Earth project has significantly contributed to this research to provide essential biological materials. I think that 1400 microorganisms depend on their project, which is indispensable material. My understanding is that this project is an educational project for undergraduate students. For their contribution and motivation in the future, I would suggest that you consider publishing them as a member of authors under the project name instead of individual names. It may have a positive impact on their future activities, I believe.

2) In Waksman's original method, the target bacteria, in this case, Escherichia coli and Bacillus subtilis, are coexisting as an indicator strain. In the mCPT method, the antibiotic production capacity was analyzed with a co-inoculating of these two bacteria. The authors tried to optimize the screening method combining with Medium conditions and negative selection using D-Alanine to improve the original CPT method and established it as an mCPT method. 

It would be quite valuable to identify which microorganisms caused to activate the antibiotic production of the antibiotic-producing bacteria as an activator strain by species interaction.

If the analysis leads to identifying the species causing antibiotic production by species interaction, I believe that this paper will stand out.

3) Line 156~158, "random assortment of 210 microbes" are culturable species? Also, the number of isolation over 1400 on line 29 is culturable? My understanding is that the initial pool of bacterial species is directly from the soil sample. After the screening with a high density of mixed culture method, further purification was performed. Is this correct? Could you describe more clearly in the materials and methods section? 

4) Related on the 2), after the first screening, which is crowded and still keeps the inter-species interaction, however, for the further screening, the authors tried to purify the target organisms and no more interaction to activate the expression of the antibiotic biosynthetic pathway genes, I think. However, those microbes kept the antimicrobial activities. So I think those screened microbes are culturable, I suppose. Is this correct? My understanding of the authors' purpose of this analysis was to develop new methods to extend screening under the condition of keeping microbial inter-species interaction to keep the expression of the antibiotic synthetic pathway genes. 

5) Table 2, 1D NMR data, I think the major purpose of this analysis is not to focus on the structure of the functional chemical compounds so that this table can be moved to the supplementary information. But, of course, this type of data is important for structural interest. 

6) line 436, there is no explanation of what GNPS is an abbreviation for.

7) Line 557, section 4.7, "16. S rRNA" is a mistake for "16S rRNA".
